



# Coupling of organic and inorganic aerosol systems and the effect on gas-particle partitioning in the southeastern United States

Havala O. T. Pye[1], Andreas Zuend[2], Juliane L. Fry[3], Gabriel Isaacman-VanWertz[4,5], Shannon L. Capps[6], K. Wyat Appel[1], Hosein Foroutan[1], Lu Xu[7], Nga L. Ng[8,9], and Allen H. Goldstein[5,10]

[1]National Exposure Research Laboratory, U.S. Environmental Protection Agency, Research Triangle Park, North Carolina, USA
[2]Department of Atmospheric and Oceanic Sciences, McGill University, Montreal, Québec, CAN
[3]Department of Chemistry, Reed College, Portland, Oregon, USA
[4]Department of Civil and Environmental Engineering, Virginia Polytechnic Institute and State University, Blacksburg, Virginia, USA
[5]Department of Environmental Science, Policy, and Management, University of California, Berkeley, California, USA
[6]Civil, Architectural, and Environmental Engineering, Drexel University, Philadelphia, Pennsylvania, USA
[7]Department of Environmental Science and Engineering, California Institute of Technology, Pasadena, California, USA
[8]School of Chemical and Biomolecular Engineering, Georgia Institute of Technology, Atlanta, GA, USA
[9]School of Earth and Atmospheric Sciences, Georgia Institute of Technology, Atlanta, Georgia, USA
[10]Department of Civil and Environmental Engineering, University of California, Berkeley, California, USA

*Correspondence to:* Havala O. T. Pye (pye.havala@epa.gov)

**Abstract.** Several models were used to describe the partitioning of ammonia, water, and organic compounds between the gas and particle phase for conditions in the southeastern United States during summer 2013. Existing equilibrium models and frameworks were found to be sufficient although additional improvements in terms of estimating pure-species vapor pressures are needed. Thermodynamic model predictions were consistent, to first order, with a molar ratio of ammonium to sulfate of approximately 1.6 to 1.8 (Ratio of ammonium to 2 × sulfate, $R_{N/2S} \approx 0.8$ to 0.9) with approximately 70% of total ammonia and ammonium ($NH_x$) in the particle. Southeastern Aerosol Research and Characterization (SEARCH) network gas and aerosol and Southern Oxidant and Aerosol Study (SOAS) Monitor for Aerosols and Gases in Air (MARGA) aerosol measurements were consistent with these conditions. CMAQv5.2 regional chemical transport model predictions did not reflect these conditions due to biases in the nonvolatile cations that resulted from either overestimated emissions and/or underestimated mixing. In addition, gas-phase ammonia was overestimated in the CMAQ model leading to an even lower fraction of total ammonia in the particle. Chemical Speciation Network (CSN) and Aerosol Mass Spectrometer (AMS) measurements indicated less ammonium per sulfate than SEARCH and MARGA measurements and were inconsistent with thermodynamic model predictions. Organic compounds were predicted to be present to some extent in the same phase as inorganic constituents, modifying their activity and resulting in a decrease in $[H^+]_{air}$ ($H^+$ in $\mu g\ m^{-3}$ air), increase in ammonia partitioning to the gas phase, and increase in pH compared to complete organic vs. inorganic liquid-liquid phase separation. In addition, accounting for non-ideal mixing modified the pH such that a fully interactive inorganic-organic system had a pH roughly 0.7 units higher than predicted by traditional methods (pH= 1.5 vs 0.7). Particle-phase interactions of organic and inorganic compounds were found to increase





partitioning towards the particle phase (vs. gas phase) for highly oxygenated (O:C≥0.6) compounds including several isoprene-derived tracers as well as levoglucosan, but decrease particle-phase partitioning for low O:C, monoterpene-derived species.

# 1 Introduction

Ambient particles consist of organic and inorganic compounds. The organic compounds present in the gas and particle phase are diverse and numerous (Goldstein and Galbally, 2007), ranging from relatively unoxidized, long-chain alkanes in fresh emissions to small, highly soluble compounds formed through multiple generations of atmospheric chemistry. Major inorganic constituents include water, sulfate, ammonium, and nitrate with additional contributions from species such as calcium, potassium, magnesium, sodium, and chloride (Reff et al., 2009). The extent to which organic and inorganic components of particulate matter interact within a particle depends on the mixing state (e.g. internal vs external) of the aerosol population as well as degree of phase separation (or number of phases) within the particle. Internally mixed populations, as typically assumed in chemical transport models such as the Community Multiscale Air Quality (CMAQ) model, may exhibit one fairly homogeneous liquid phase state or be heterogeneous in composition. Heterogeneous configurations occur as a result of phase separation and may include a liquid and solid phase or multiple liquid phases. A common heterogeneous configuration under conditions of liquid-liquid or solid-liquid phase separation is that of a core-shell morphology; alternatively, partially engulfed morphologies have been predicted by theory and observed in laboratory experiments (Kwamena et al., 2010; Song et al., 2013; Reid et al., 2011).

Currently, the CMAQ model, as well as other chemical transport models, considers accumulation mode aerosol to form a heterogeneous internal mixture in which organic and inorganic constituents partition between the gas and aerosol phase independently of each other. Pye et al. (2017) examined how assumptions about phase separation of internally mixed particles affect organic aerosol concentrations in the southeast United States as predicted by the CMAQ model. When organic compounds were allowed to mix with the aqueous inorganic phase under conditions of high relative humidity and high degree of oxygenation (You et al., 2013; Bertram et al., 2011; Song et al., 2012), the concentration of organic aerosol was predicted to increase significantly (Pye et al., 2017). While the effects of phase separation on organic compounds are potentially large, they are highly dependent on an accurate parameterization of activity coefficients and a reliable prediction of the composition of individual particle phases (Zuend and Seinfeld, 2012).

Recent work highlights potential discrepancies between current gas-particle partitioning models, which assume equilibrium is attained on short timescales, and observations for both inorganic and organic compounds. Silvern et al. (2017) found that models predict higher ratios of particulate ammonium to sulfate than observed in the Eastern U.S. and proposed that organic compounds in an organic-rich phase at the particle surface may reduce ammonia partitioning to the particle via a kinetic inhibition. In addition, organic compound vapor pressure estimation method predictions can vary by orders of magnitude (Topping et al., 2016; O'Meara et al., 2014; Pankow and Asher, 2008) and have often been adjusted downward to improve model predictions (Chan et al., 2009; Johnson et al., 2006). Futhermore, isoprene-epoxydiol-derived organic aerosol partitions to the particle phase to a greater degree than structure-based vapor pressures would suggest (Isaacman-VanWertz et al., 2016;



Lopez-Hilfiker et al., 2016; Hu et al., 2016). Since $PM_{2.5}$ (particulate matter concentration from particles of diameters less than 2.5 $\mu$m) is regulated via the National Ambient Air Quality Standards (NAAQS) in the U.S., while similar ambient standards are not set for the gas-phase counterparts, errors in partitioning will affect model performance with implications for metrics used in regulatory applications. The model sensitivity of $PM_{2.5}$ to emission changes can also be too high or too low if compounds

are erroneously partitioned.

In this work, gas-particle partitioning of ammonia and several isoprene-, monoterpene-, and biomass burning-derived organic compounds was examined using common air quality modeling treatments and advanced approaches. Results address the degree to which techniques accounting for organic-inorganic interactions, deviations in ideality, and phase separation reproduce observations. Models were evaluated for their ability to predict ammonia versus ammonium as well as gas-particle partitioning

of organic compounds. In addition, the effects of organic compounds on aerosol pH were examined.

## 2   Methods

### 2.1   Model Approaches

Several box-model approaches as well as CMAQ regional chemical transport model calculations were used to represent the partitioning of compounds between the gas and particle phases. CMAQ version 5.2-gamma was run over the continental U.S.

at 12 km by 12 km horizontal resolution for 1 June – 15 July 2013, coinciding with the Southern Oxidant and Aerosol Study (SOAS) field campaign and the Centreville, Alabama, U.S. field site. WRF v3.8 meteorology with lightning assimilated into the convection scheme (Heath et al., 2016) was processed for use with the CMAQ model (Otte and Pleim, 2010). Emissions were based on the 2011 National Emission Inventory version 2 and year 2013 specific when available. Windblown dust emissions followed the scheme of Foroutan et al. (2017). Ammonia emissions and deposition from croplands were parameterized as

a bidirectional exchange (Pleim et al., 2013). CMAQ used ISORROPIA v2.1 (Fountoukis and Nenes, 2007; Nenes et al., 1998) with gas and aerosol composition and environmental conditions (temperature, relative humidity) as input to predict the Aitken and Accumulation mode ammonium, nitrate, and chloride mass concentrations. CMAQ predicted $PM_1$ and $PM_{2.5}$ were computed based on the fraction of the Aitken and Accumulation modes less than 1 or 2.5 microns in diameter as appropriate (Nolte et al., 2015).

Consistent with the CMAQ regional model, partitioning of ammonia between the gas and particle phases was also predicted using ISORROPIA as a box model driven with observed aerosol or gas and aerosol concentrations of ammonia, ammonium, nitrate, nitric acid, calcium, potassium, magnesium, sodium, and chloride. Output from the ISORROPIA box-model was either gas-phase ammonia in equilibrium with the observed aerosol ammonium, or ammonia vs. ammonium based on total gas and aerosol conditions. ISORROPIA does not consider the effects of organic compounds on aerosol pH or explicitly treat liquid-

liquid phase separation.

Algorithms that allowed for inorganic-organic interactions were applied using a thermodynamic equilibrium gas-particle partitioning model (Zuend et al., 2010; Zuend and Seinfeld, 2012) based on the Aerosol Inorganic-Organic Mixtures Functional groups Activity Coefficients (AIOMFAC) model (Zuend et al., 2008, 2011). AIOMFAC provided an estimate of activity





coefficients for aerosol systems of specified functional group composition, which was used in two modes: (i) predefined complete liquid-liquid phase separation (CLLPS) in which the organic compounds did not mix with the inorganic salts and (ii) equilibrium (EQLB) in which the Gibbs energy of the system was minimized and up to two liquid phases of any composition were allowed to form in the particle as predicted by a modified liquid-liquid phase separation algorithm based on the method

by Zuend and Seinfeld (2013). For purposes of AIOMFAC calculations, observed calcium, potassium, and magnesium concentrations were converted to charge-equivalent sodium amounts since the former's interactions with the bisulfate ion in solution are not treated by the model.

Several quantities, including pH and molar ratios, were calculated to evaluate the inorganic aerosol system. Solution acidity can be expressed in different ways, the most common one being the pH value. However, many definitions of pH exist, with

10 several definitions only applicable to highly dilute aqueous solutions. Thermodynamics-based pH definitions vary with the choice of composition scale (molality, molarity, or mole fraction basis) and the solvent into which $H^+$ is assumed to dissolve, which may be strictly water associated with inorganic constituents as in ISORROPIA II, or include the diluting effect of water associated with organic compounds (Guo et al., 2015), organic compounds themselves (Zuend et al., 2008), or other aerosol constituents (Budisulistiorini et al., 2017). Furthermore activity coefficients of $H^+$ may not be unity as is frequently

assumed. In this work, pH was defined following the thermodynamic definition on a molality basis, as recommended by IUPAC (http://goldbook.iupac.org/html/P/P04524.html) and computed by the AIOMFAC model. By expressing the molality of $H^+$ in terms of concentration per volume of air, the following results:

$$pH = -\log_{10}(\gamma_{H^+}[\text{H}^+]_{\text{air}}/[S]) \qquad (1)$$

where $\gamma_{H^+}$ is the molality based activity coefficient for $H^+$ in the liquid phase, $[\text{H}^+]_{\text{air}}$ is the concentration of the hydronium

ion in the liquid phase in moles per volume of air and $[S]$ is the solvent mass in that liquid phase (kg per volume of air), i.e. $[\text{H}^+]_{\text{air}}/[S]$ is the molality of $H^+$. The solvent included water associated with inorganic compounds ($W_i$), water associated with organic compounds ($W_o$), and organic compounds ($C_{\text{org}}$) as appropriate based on the predicted phase composition. ISORROPIA pH calculations assumed $[S] = [W_i]$ and an activity coefficient of unity thus following previous methods (Guo et al., 2017). The molar ratio of ammonium to 2 × sulfate was defined as:

$$R_{N/2S} = \frac{n_{\text{NH}_4^+}}{2 \times n_{\text{SO}_4^{2-}}}, \qquad (2)$$

and the electric charge normalized molar ratio of cations to anions that participate in ISORROPIA was:

$$R_{+/-} = \frac{n_{\text{NH}_4^+} + n_{\text{Na}^+} + 2 \times n_{\text{Ca}^{2+}} + n_{\text{K}^+} + 2 \times n_{\text{Mg}^{2+}}}{2 \times n_{\text{SO}_4^{2-}} + n_{\text{NO}_3^-} + n_{\text{Cl}^-}}. \qquad (3)$$

Since ambient measurements and CMAQ model output do not distinguish bisulfate from sulfate, the sulfate in these ratios represented total sulfate ($SO_4^{2-} + HSO_4^-$).

To employ the AIOMFAC-based equilibrium models, organic aerosol positive matrix factorization (PMF) analysis results of ambient data (next section) were converted to molecular structures of known functional group composition as surrogates for a range of organic compound classes in ambient particles as described in Tables S1-S3 thus providing a complete characterization





of the organic aerosol partitioning medium. Several isoprene-derived (2-methyltetrols, $C_5$-alkene triols, 2-methylglyceric acid) and monoterpene-derived (pinic acid, pinonic acid, hydroxyglutaric acid) compounds as well as levoglucosan, a semivolatile indicator of biomass burning, were explicitly represented in box-model calculations. Pure species' vapor pressures (sub-cooled liquid) were obtained via the EVAPORATION model (Compernolle et al., 2011). The temperature dependence was parame-

5 terized by using the same Antoine-like function that is also employed by the EVAPORATION model. A sensitivity calculation (referred to as Adj Psat) reduced EVAPORATION-based vapor pressures by a factor of 4.2, thus maintaining the compound to compound variability predicted by EVAPORATION but correcting for potential overestimates in pure compound vapor pressures. The magnitude of the adjustment was based on the effective saturation concentration obtained via regression needed to reproduce observations in a traditional absorptive partitioning framework (Equation S1). This adjustment factor is similar

in magnitude to the difference between SIMPOL (Pankow and Asher, 2008) and EVAPORATION (Compernolle et al., 2011) predicted vapor pressures for several species, but not all (see Table S4). The effective saturation concentration, C*, of a species, i, was defined as (Zuend and Seinfeld, 2012):

$$C_i^* = \frac{C_i^g \sum C_k^{PM}}{C_i^{PM}} \qquad (4)$$

where $C_i^g$ is the mass-based gas-phase concentration of species $i$, $C_i^{PM}$ is the mass-based liquid-phase concentration of species

$i$, and $C_k^{PM}$ is the total mass-based concentration of the liquid phase where the summation index $k$ includes organic species, inorganic species, and water. See Equation S2 for $C_i^*$ in terms of the mole-fraction based activity coefficient.

## 2.2 Ambient Data

Regional model predictions of inorganic aerosol were evaluated against the Chemical Speciation Network (CSN) and Southeastern Aerosol Research and Characterization (SEARCH) network observations (at different ground sites). CSN determines

anions and cations via ion chromotography of extracts from nylon filters (Solomon et al., 2014). The SEARCH network uses a multichannel approach employing nylon, teflon, and citric acid-coasted cellulose filters to measure speciated 24-hour average $PM_{2.5}$ (Edgerton et al., 2005). The SEARCH 24-hour filter measurements are also used to adjust the co-located continuous measurements (Edgerton et al., 2006).

In addition to the network data, ambient data from SOAS at the Centreville, AL (CTR; 87.25° W, 32.90° N) site from June

and July, 2013 were used as input to the box models and for model evaluation. The High Resolution Time of Flight Aerosol Mass Spectrometer (HR-ToF-AMS, hereafter AMS) operated by the Georgia Institute of Technology was the primary source of SOAS $PM_1$ organic mass, ammonium, and sulfate (Xu et al., 2015a). When AMS data was used as input in $PM_1$ box modeling, inorganic nitrate was set to zero as nitrate measured by the AMS contained significant contributions from organic nitrogen-containing compounds (Xu et al., 2015b). Thus, AMS calculations assumed the inorganic aerosol was composed only

of ammonium, sulfate, bisulfate, and the hydronium ion (referred to in subsequent sections as the A' system). The assignment of measured ammonium and sulfate to specific salts (ammonium sulfate vs ammonium bisulfate) for use as input electrolyte components to AIOMFAC was determined by mass balance. For ammonium-sulfate only conditions in which the moles $NH_4^+$ $\geq 2 \times SO_4^{2-}$, a small amount ($1 \times 10^{-4}$ $\mu$g m$^{-3}$) of ammonium bisulfate was added to the AIOMFAC input in order to trigger





a potential partial association of sulfate and $H^+$ ions to bisulfate following the equilibrium constant of that reaction. Inorganic $PM_{2.5}$, including ammonium, sulfate, nitrate, calcium, potassium, magnesium, sodium, and chloride, was measured at CTR by a Monitor for Aerosols and Gases in Air (MARGA) (Allen et al., 2015). Less than 5% of the $PM_{2.5}$ MARGA data used in this work had elevated nitrate (>0.8 $\mu g\ m^{-3}$) due to supermicron crustal material and sea salt episodes (Allen et al., 2015). Gas-phase ammonia was obtained from the CTR SEARCH network site via a corrected Thermo Scientific citric acid-impregnated denuder. Relative humidity (RH) and temperature were obtained from the routine SEARCH network measurements at the SOAS site.

The entire organic aerosol composition was characterized in terms of functional groups for use with AIOMFAC. The semi-volatile thermal desorption aerosol gas chromatograph (SV-TAG) with in situ derivatization (Isaacman-VanWertz et al., 2016; Isaacman et al., 2014) provided measured gas- and aerosol-phase concentrations of 2-methyltetrols, $C_5$-alkene triols, 2-methylglyceric acid, pinic acid, pinonic acid, hydroxyglutaric acid, and levoglucosan. More oxidized-oxygenated organic aerosol (MO-OOA), biomass burning organic aerosol (BBOA), Isoprene-OA, and less oxidized-oxygenated organic aerosol (LO-OOA) PMF factors from the AMS were represented with specific functional groups and associated surrogate chemical structures (Table S1). As previous work indicates a fraction of measured 2-methyltetrols may be decomposition products of low-volatility accretion products (Isaacman-VanWertz et al., 2016; Lopez-Hilfiker et al., 2016), 50% (as a rough estimate) of measured 2-methyltetrols (in the particle phases) were assumed to be dimer decomposition products when EVAPORATION-based vapor pressures were used (see Table S4). In the sensitivity calculation (Adj Psat), 2-methyltetrols were assumed to be present only in the monomer form as including dimers increased the model bias.

The overlap in the input data sets resulted in 180 hours of measurement coverage. Additional measurements of ammonium, sulfate, and ammonia (not used in this work) are summarized in Tables S5-S7 for reference.

## 3 Results and Discussion

### 3.1 Regional ammonium-sulfate conditions

Figure 1 shows the molar ratios of ammonium to 2 × total sulfate and cations to anions over the eastern U.S. for 1 June - 15 July 2013 based on observations from the CSN network and predicted by CMAQv5.2. CMAQ predicted a mean $R_{N/2S}$ 0.73 over the U.S. compared to the observed mean of 0.67. The model showed higher values (near 1) over the central U.S. and lower values (<0.6) over the southeast U.S. The magnitude of the observed $R_{N/2S}$ was similar to that from the CSN network (normalized mean bias <10%). However, significant discrepancies existed between the model and observations for the ratio of cations to anions. The observations indicated that the ratio of ammonium to sulfate was a good proxy for the ratio of cations to anions. In CMAQ, however, the ratio of cations to anions was approximately one. Molar ratios are not robust indicators of aerosol pH (Hennigan et al., 2015) as a result of the role of relative humidity and associated liquid water content as well as buffering by bisulfate (Guo et al., 2015). However, chemical transport model biases in ion ratios should result in biases in acidity and gas-particle partitioning of volatile acids and bases (e.g. $NH_3$) considering other factors (such as RH) held constant.



An evaluation of the individual cations and anions (Figures S1-S5) indicates CMAQ over predicted the non-volatile ISOR-ROPIA cations ($Na^+$, $Ca^{2+}$, $K^+$, $Mg^{2+}$) which was not revealed in the $R_{N/2S}$ comparison in Figure 1. Appel et al. (2013) have previously shown that even when anthropogenic fugitive dust and windblown dust emissions are removed from the CMAQ model, crustal elements are still typically overestimated compared to observations. Coal combustion, for example, is a major

source of trace metals in the U.S. (Reff et al., 2009). Trace metal emissions were overestimated (and/or physical mixing was underestimated) since CMAQ overestimated their measured concentration, which included soluble and insoluble contributions (Solomon et al., 2014). Since ISORROPIA should only consider the cations associated with sulfate, nitrate, and chloride, but CMAQ includes cations that are part of insoluble metal oxides (Reff et al., 2009), additional error was incurred in CMAQ by allowing all of the calcium, potassium, magnesium, and sodium present in aerosol to participate in ISORROPIA calculations.

Thus, the apparent consistency in ammonium to sulfate ratios between CSN and CMAQ, should not be used to confirm the reasonableness of either. The ratio of cations to anions indicates discrepancies between CSN and CMAQ, specifically, that the CMAQ model tends to achieve charge balance as defined by $R_{+/-}=1$ while observations indicate otherwise.

Also included in Figure 1(a-b) are observations of $R_{N/2S}$ based on the SEARCH network (triangles) which are much higher (>0.8) than the CSN values (<0.6) in the southeast U.S. While there could be spatial heterogeneity in the southeast U.S.,

differences so large are unlikely and not present in CMAQ, thus indicating potential problems in one set of measurements. Nylon filters (used by CSN for inorganic ions) can collect 4-5% of gas-phase sulfur dioxide (Benner et al., 1991; Hansen et al., 1986), leading to a small but positive sulfate mass concentration artifact. In addition, nylon filters tend to measure lower ammonium concentrations than other filter types (Solomon et al., 2000; Yu et al., 2006). These ammonium artifacts are not restricted to ammonium nitrate since more than twice as much $NH_4^+$ was lost compared to nitrate on nylon filters from Great

Smoky Mountains National Park, TN, U.S. (Yu et al., 2006). 6-14% of total $NH_4^+$ can volatilize in federal reference method (FRM) collection, and the SEARCH network best estimates of $PM_{2.5}$ result in higher ammonium on an absolute basis and as a fractional contribution to $PM_{2.5}$ compared to the FRM equivalent mass (Edgerton et al., 2005). Consider that a 10% underestimate in ammonium PM and 10% overestimate in sulfate will lead to almost a 20% underestimate in $R_{N/2S}$.

### 3.2 Ammonia gas-particle partitioning during SOAS

Consistent with CMAQ predictions over the greater southeastern U.S. region, CMAQ predicted an average ratio of ammonium to 2×sulfate ($R_{N/2S}$) of 0.64-0.61 (for $PM_1$ and $PM_{2.5}$ respectively) and 24-28% of total ammonia in the particle as ammonium (b and g in Figure 2) at CTR. CMAQ also predicted that the cation to anion charge ratio, $R_{+/-}$, was near one during SOAS. Thus, CMAQ predictions for SOAS CTR site were representative of the southeastern United States for further investigating CMAQ model issues related to inorganic molar ratios and ammonia partitioning.

As shown in Figure 2, the CMAQ predicted $R_{N/2S}$ (b) was similar to the Georgia Tech AMS derived value (a). It was also similar to the regional SEAC$^4$RS AMS-derived values (Silvern et al., 2017) (Table S6) which averaged near 0.6. ISORROPIA predictions using AMS measured ammonium and sulfate as input (c) showed much higher partitioning of ammonia to the particle phase (mean $NH_x$ $F_p$ of 0.8) than indicated by AMS aerosol data combined with SEARCH ammonia. Using total ammonia and ammonium as model input resulted in a similar fraction of $NH_x$ in the particle as using only ammonium as input,





but the $R_{N/2S}$ value significantly increased to around 0.8 (d). The AIOMFAC-based equilibrium model run with aerosol-only inputs (e) was qualitatively consistent with ISORROPIA (c) in terms of the fraction of $NH_x$ in the particle. These tests indicated that AMS measurements at SOAS CTR were inconsistent with ISORROPIA and AIOMFAC thermodynamic calculations, as indicated in previous model evaluation (Silvern et al., 2017). This behavior could have resulted from a deviation from gas-

particle equilibrium, missing thermodynamic effects, or biases in the AMS data. Given that relative humidities are high in the eastern US during summer such that particles should not have diffusivity limitations on a one-hour timescale (Renbaum-Wolff et al., 2013), this work examines the latter two possibilities with an emphasis on the potential bias in AMS data.

The $R_{N/2S}$ determined from the MARGA instrument for $PM_{2.5}$ (f) was significantly higher than that derived from the AMS measurements and closer to the values based on SEARCH measurements (Table S5, Figure 1). The AMS tended to measure

much less ammonium and the same or slightly less sulfate than the MARGA thus biasing the AMS determined $R_{N/2S}$ low (Table S5-S6). As a result, the fraction of ammonia partitioned to the particle using SEARCH $NH_3$ and MARGA aerosol measurements was higher than would be estimated using AMS data. The CMAQ model calculations showed a small but similar trend as observations for $PM_1$ to $PM_{2.5}$ in terms of ammonia gas-particle partitioning (since $PM_{2.5} \geq PM_1$ and $F_p =$ PM / ( PM+gas ) ) but did not show significantly increased $R_{N/2S}$ with increased particle size.

ISORROPIA $PM_{2.5}$ calculations using both gas and aerosol inputs were run with (i in Figure 2) and without (j) aerosol calcium, potassium, magnesium, sodium, nitrate, and chloride and the results were qualitatively the same in terms of mean fraction of ammonia partitioned to the particle and ratio of $NH_4^+$ to sulfate in the particle. Thus, the difference between $PM_1$ and $PM_{2.5}$ was primarily driven by the difference in ammonium and sulfate measured by the AMS versus MARGA instrument, not the difference in size of particles sampled or the availability of nonvolatile cations. Overall, ISORROPIA and AIOMFAC

were qualitatively consistent with MARGA measurements of $R_{N/2S}$, but not with AMS measurements. Note that in full CMAQ model calculations, the nonvolatile cations were so high that they erroneously affected the partitioning of ammonium.

The differences in the AMS and MARGA datasets in terms of $R_{N/2S}$ are consistent with a potential bias in AMS collection efficiency (CE) for ammonium sulfate vs ammonium bisulfate and/or the presence of organosulfates in AMS measured sulfate. AMS instruments are known to have a higher collection efficiency for acidic ($H_2SO_4$-enriched) vs $(NH_4)_2SO_4$-enriched

aerosol and a composition dependent collection efficiency is applied to ambient data (Middlebrook et al., 2012). If there was a bias in the calculated CE as a function of $R_{N/2S}$, it would lead to a bias in measured $R_{N/2S}$. The magnitude of this potential bias is not clear. If the difference in AMS vs MARGA data was a result of measurement of $PM_1$ by the AMS vs $PM_{2.5}$ by MARGA, then AMS measured ammonium and sulfate concentrations should always be lower than $PM_{2.5}$ measurements, which is not the case. Guo et al. (2015) compared Georgia Tech AMS $PM_1$ measurements to the Particle into Liquid Sampler

(PILS) $PM_{2.5}$ measurements and showed that AMS sulfate was high (by 20%) compared to PILS $PM_{2.5}$ measured sulfate, implying the AMS has a low $R_{N/2S}$. A portion of the AMS overestimate in sulfate (relative to ammonium) may be due to organosulfates which accounted for roughly 5% of measured sulfate during SOAS (Hu et al., 2017).



### 3.3 Phase composition

Figure 3 shows the average concentration of aerosol components predicted in the electrolyte-rich ($\alpha$) and organic-rich ($\beta$) aerosol phases as well as under conditions in which only one liquid phase was predicted (single phase) based on AIOMFAC equilibrium calculations (EQLB) of the aqueous ammonium-sodium-sulfate-nitrate-chloride (A) and organic surrogates system. In all cases, water was predicted to be a major contributor to the phase accounting for 60%, 35%, and 90% of the mass in the average $\alpha$, $\beta$, and single phases respectively. In addition, inorganic ions were present in all phases including the organic-rich phase. This means that the effects of inorganic species on organic compounds were not limited to times when one single liquid phase was predicted. Higher concentrations of organic species were generally associated with an increase in the predicted frequency of phase separation. However, LO-OOA, the least oxygenated (Table S2) and least water-soluble secondary organic aerosol PMF factor (Xu et al., 2017), was not more or less abundant when phase separated vs. single phase conditions were predicted.

The mean $R_{N/2S}$ varied slightly by phase with the $\alpha$ phase having a value of 0.8 and the phases with a greater proportion of organic compounds ($\beta$ and single) having a value of 0.9. The $\beta$ phase, with its high concentration of organic species, showed a lower $[H^+]_{air}$ compared to the $\alpha$ phase. The ammonium-sulfate only (in terms of inorganic ion representation) system was predicted to have the same frequency of phase separation and trend in $[H^+]_{air}$, but less difference in the $R_{N/2S}$ between the phases.

Phase separation into electrolyte-rich and organic-rich phases was predicted to occur 70% of the time. The frequency of phase separation predicted for SOAS conditions was higher than the frequency predicted in previous CMAQ work (Pye et al., 2017) that calculated separation relative humidities based on average O:C ratios using the parameterization of You et al. (2013) for a particular inorganic salt type. Both the previous CMAQ calculations (Pye et al., 2017) and this work predicted the same diurnal variation with a greater frequency of phase separation during the day driven by lower relative humidities (Figure S9).

### 3.4 Effects of organic compounds on acidity

Acidity (pH) is an important aerosol property as it promotes dissolution of metals (Fang et al., 2017), increases nutrient availability (Stockdale et al., 2016), and catalyzes particle-phase reactions (Eddingsaas et al., 2010). Current recommended methods for estimating aerosol pH include thermodynamic models and ammonia-ammonium partitioning (Hennigan et al., 2015) as direct measurements are difficult to make (Rindelaub et al., 2016). AIOMFAC predicted a median molal pH of 1.4 (ammonium-sulfate system) to 1.5 (ammonium-sodium-sulfate-nitrate-chloride system) for SOAS conditions (Table 1). AIOMFAC occasionally showed high pH (pH = 7, Figure 4) which occurred when an excess of cations compared to anions were observed, leading to the absence of $H^+$ and bisulfate in the input compositions used with the model. Similar behavior has occurred with ISORROPIA and the AIM thermodynamic models using aerosol-only inputs (Hennigan et al., 2015) and likely resulted from measurement uncertainty and a resulting high-bias in the measured amounts of cations compared to charge-equivalent anions. The ISORROPIA predicted pH for the subset of conditions used here (pH = 0.7 to 1.1) was similar to those





previously reported for SOAS (pH = 0.9) and Southeast Nexus (SENEX) aircraft campaign (pH = 1.1) using other datasets as summarized by Guo et al. (2017).

Regardless of whether only ammonium-sulfate or ammonium-sodium-sulfate-nitrate-chloride sytems were treated, AIOM-FAC predicted an increase in the concentration of gas-phase ammonia (decrease in $NH_x$ $F_p$) along with a decrease in acidity when organic compounds were considered in the calculation of partitioning (EQLB vs CLLPS, Table 1, Figure 4). The presence of organic compounds in the same phase as $H^+$ and other ions (EQLB case) shifted free $H^+$ towards increased association with sulfate to form bisulfate as AIOMFAC predicts bisulfate to be more miscible with organic compounds than $H^+$ and other small cations. Interactions with organic compounds resulted in a 34-36% decrease in median $[H^+]_{air}$ and a 0.1 unit increase (11-12% increase) in median pH.

If the pH for forced complete phase separation conditions was recalculated using AIOMFAC CLLPS predicted $[H^+]_{air}$ and assuming an activity coefficient of one (traditional method), the resulting pH has a median of 0.7 (Figure 4b), the same value obtained by ISORROPIA using only aerosol inputs and an activity coefficent of unity. Thus, traditional methods resulted in an artificially low pH. Taking into account activity coefficients other than unity, phase separation, and the diluting effect of organic compounds and their associated water (EQLB) resulted in a pH 0.7 pH units higher than traditional methods. This increase is substantial given that the pH scale is logarithmic; a 0.7 pH unit higher value is equivalent to a five times lower molal $H^+$ activity in solution. The activity coefficient value was a major driver of this difference with a secondary role for solvent abundance and change in $[H^+]_{air}$.

## 3.5 Partitioning of organic compounds under ambient conditions

For organic compounds with O:C≥0.6 ($C_5$-alkene triols, levoglucosan, 2-methyltetrols, hydroxyglutaric acid, 2-methylglyceric acid), the particle-phase fraction, $F_p$, was predicted to increase when the electrolyte-rich and organic-rich phases were allowed to equilibrate (EQLB compared to CLLPS, Figures 5-6) as a result of an increase in the abundance of the partitioning medium. For compounds with lower O:C (specifically pinic and pinonic acid) $F_p$ decreased as a result of unfavorable liquid-phase interactions. The increase in $F_p$ for most species generally resulted in a decrease in the mean bias and mean error of $F_p$ compared to observations (Figure 5b). With the pure-species adjusted vapor pressure (Adj Psat sensitivity), the mean bias in $F_p$ for all organic species was less than 0.2 and emphasized that information about the pure species vapor pressure is important for accurate gas-particle partitioning calculations. The influence of inorganic constituents on organic compound partitioning was not limited to the times when one single phase was present. In the case of hydroxyglutaric acid (Figure 6g), predictions of $F_p$ were found to be most sensitive to assumptions regarding condensed phase mixing during the day when phase separation was most common (coinciding with a lower average RH during midday and afternoon hours, as expected). This occurred because the organic-rich phase still contained a significant amount of inorganic ions (Figure 3) which modified the partitioning medium and impacted the predicted activity of the organic species.

The change in $F_p$ between CLLPS and EQLB calculations was consistent with the change in effective saturation concentrations (Figure 5c). The effective C* (equation 4) under equilibrium (EQLB) conditions compared to CLLPS (EQLB C*/CLLPS C*) was a strong function of the compound O:C ratio (Pearson's $r^2$=0.79) with higher O:C species having lower



EQLB C*/CLLPS C* ratios. The mean activity coefficient value was predicted to either stay the same (2-methylglyceric acid) or increase (all other explicit organic species) in EQLB compared to CLLPS. Thus, the driving factor for increased partitioning to the particle phase (indicated by increased $F_p$ and decreased C*) for species with O:C>0.6 under EQLB compared to CLLPS was the ability of the increased partitioning medium size to overcome the increased activity coefficients. The increased

partitioning medium gained by interacting with the inorganic species and their water lowered the mole fraction of the organic species in the particle, thus leading to lower predicted particle-phase activity and gas-phase concentrations via modified Raoult's law. In some cases, like for 2-methyltetrols, the species exhibited negative deviations from ideality ($\gamma$ <1) in both CLLPS and EQLB, but the activity coefficient still increased from CLLPS to EQLB (Table S9). For pinic and pinonic acid, the deviation ($\gamma$ >1) was positive in CLLPS and its activity coefficient even larger in magnitude in EQLB such that the larger

partitioning medium did not overcome the deviation in ideality resulting in the species being more abundant in the gas phase in EQLB compared to CLLPS. Interestingly, levoglucosan was the only species predicted to have an activity coefficient near 1 for the organic-rich ($\beta$) phase in EQLB calculations (Table S9). Due to the effect on the size of the partitioning medium resulting from additional species (specifically water and inorganic salts) in the $\beta$ phase during EQLB, the effective C* for levoglucosan was predicted to be 35% of its pure species value (1.4 $\mu$g m$^{-3}$, Table S8).

Predicted unfavorable interactions (limited miscibility within both the organic-rich and inorganic-rich liquid phases) resulted in pinonic acid (Figure 6f) being partitioned to the gas phase to a much greater degree than the measurements indicated. Model performance was consistent with previous work in which multiple measurement techniques showed slightly higher $F_p$ than model predictions (Thompson et al., 2017). Formation of a second organic-rich phase (a third liquid phase) containing lower O:C compounds, which was not allowed in the AIOMFAC calculations, could improve pinonic acid partitioning predic-

tions. The lack of a resolved hydrocarbon-like organic aerosol (HOA) component (Xu et al., 2015a) and representation of its associated functional groups in the model may have also contributed to an unfavorable environment for low O:C compounds.

Overall, the treatment of liquid phase mixing vs separation did not improve the mean bias in 2-methyltetrol predicted $F_p$. It also did not significantly change the mean error. The average fraction of 2-methyltetrols in the particles was represented fairly well by assuming half of the measured 2-methyltetrols are actually decomposition products of a fairly nonvolatile (C*=$10^{-6}$

$\mu$g m$^{-3}$) dimer compound (dark grey square, Figure 5a,b). However, this assumption did not perform equally well at all times of day. Figure 6a indicates that the 50% dimer assumption leads to an underestimate in 2-methyltetrol $F_p$ during the day and overestimate at night. Modeling 2-methyltetrols as entirely monomers with a pure species C* of 8 $\mu$g m$^{-3}$ at 298.15K (factor of 4.2 reduction in EVAPORATION predicted vapor pressure) reproduced the daytime 2-methyltetrol partitioning well, but overestimated partitioning to the particle at night. Even with the reduced P$^{sat}$ (in the Adj Psat sensitivity), 2-methytetrol

monomers remained slightly more volatile than predicted by SIMPOL (C*=5 $\mu$g m$^{-3}$) at reference conditions. The average effective 2-methyltetrol C* (accounting for the effects of temperature and partitioning medium) in the case of CLLPS was 6 $\mu$g m$^{-3}$ while in the equilibrium calculations (EQLB) it was reduced further to 3.7 $\mu$g m$^{-3}$ (Table S8). Thus, 2-methyltetrols behaved like compounds with an effective mean saturation concentration roughly half of the pure species value due to the influence of temperature and presence of other species in the particle.



## 4 Conclusions

In this work, conditions over the eastern United States were examined with a focus on gas-particle partitioning during the Southern Oxidant and Aerosol Study (SOAS). Different measurement techniques indicated fairly different ratios of ammonium to 2 × total sulfate with the AMS instruments having the lowest values followed by CSN. The MARGA instrument (Allen

et al., 2015) and SEARCH network indicated the highest ratios of ammonium to 2 × sulfate of slightly less than 1 (mean of 0.8 to 0.9). Thermodynamic equilibrium models (ISORROPIA and AIOMFAC) are consistent with high ammonium to 2 × sulfate ratios in conjunction with about 70 to 80% of ammonia in the particle. Lower ammonium to sulfate ratios imply much higher fractions of total ammonia in the particle as thermodynamic equilibrium assumptions (and models) generally do not allow a large excess of gas-phase ammonia under highly acidic conditions. To improve CMAQ predictions of ammonium

to sulfate ratios and, therefore, pH needed for other aerosol processes such as isoprene-epoxydiol uptake (Pye et al., 2013), model predictions of nonvolatile cations should be improved in conjunction with ammonia emissions. While consideration of inorganics mixing in liquid phases with organic compounds may increase pH significantly compared to estimates from traditional models like ISORROPIA, that effect is likely not the cause of current inorganic aerosol model evaluation issues.

AIOMFAC-based predictions of gas-particle partitioning of organic compounds were sensitive to pure species vapor pres-

15 sure estimates and predictions generally had a lower mean bias when EVAPORATION-based vapor pressures were adjusted downward by a factor of 4.2 and close to values estimated by SIMPOL for 2-methyltetrols, pinic acid, and hydroxyglutaric acid. AIOMFAC predicted organic compounds interact with significant amounts of water and inorganic ions. 2-methyltetrol predictions had roughly the same error in particle fraction ($F_p$) assuming 50% of measured particulate 2-methyltetrols were decomposition products or if their vapor pressure was adjusted downward by a factor of 4.2 (to $P^{sat}=1.4\times10^{-4}$ Pa at 298.15

20 K).

*Code and data availability.* CMAQ model code is available at https://github.com/USEPA/CMAQ and v5.2-gamma was used in this work.

ISORROPIA is available from http://isorropia.eas.gatech.edu/.

AIOMFAC can be run online (http://www.aiomfac.caltech.edu/) or via contact with A. Zuend.

SOAS data is available at https://esrl.noaa.gov/csd/groups/csd7/measurements/2013senex/.

CSN data is available at https://www.epa.gov/outdoor-air-quality-data.

Model output associated with the final article will be available from the EPA Environmental Dataset Gateway at https://edg.epa.gov/ if accepted.

*Competing interests.* The authors declare no competing interests.



*Disclaimer.* The US EPA through its Office of Research and Development supported the research described here. It has been subjected to Agency administrative review and approved for publication but may not necessarily reflect official Agency policy.

*Acknowledgements.* We thank J. Jimenez, G. Ruggeri, S. Takahama, and S. Lee for providing additional datasets that are summarized in the supporting information. We thank the CSN and SEARCH networks for providing long-term measurements. We thank two reviewers at EPA.

5 We thank Paul Solomon for useful discussion. We thank CSRA for preparing emissions and meteorology input for CMAQ simulations. AZ was supported by the Natural Sciences and Engineering Research Council of Canada (NSERC), grant RGPIN/04315-2014. JLF acknowledges support from EPA-STAR RD-83539901. GIVW was supported by the NSF Graduate Research Fellowship (DGE 1106400). Fp of organic compounds collected by SV-TAG at SOAS was supported by grants to UC Berkeley including NSF Atmospheric Chemistry Program 1250569 and Department of Energy SBIR grant DE-SC0004698. LX and NLN acknowledge support from National Science Foundation

10 (NSF) Grant 1242258 and US Environmental Protection Agency (EPA) STAR Grant RD-83540301.





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





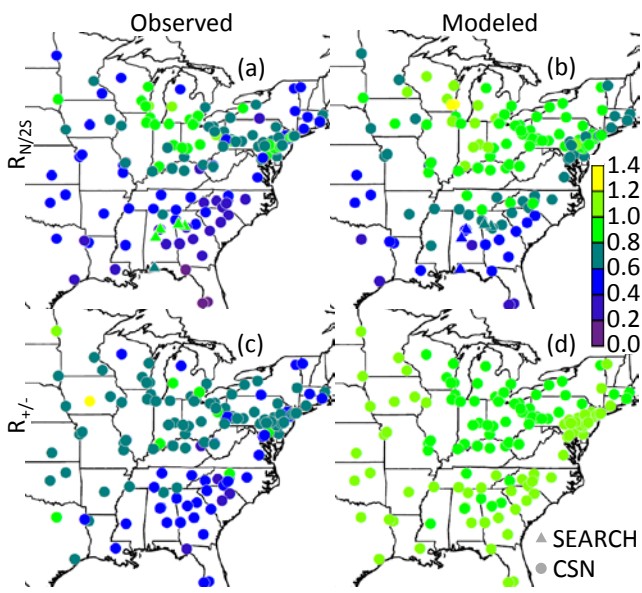

**Figure 1.** Molar ratio of aerosol ammonium to 2×sulfate ($R_{N/2S}$) (a-b) and cations to anions ($R_{+/-}$) (c-d) over the eastern US for June 1- July 15, 2013 based on observations and predicted by the CMAQ model.





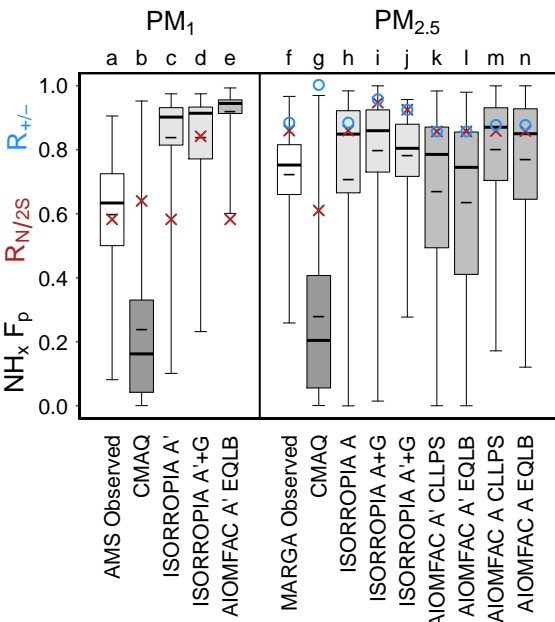

**Figure 2.** Gas-particle partitioning of ammonia ($NH_x$ $F_p$ = ammonium/(ammonia+ammonium)), mean $R_{N/2S}$ (red ×), and mean $R_{+/-}$ (blue ○) for $PM_1$ measured by the Georgia Tech AMS (Xu et al., 2015a) and $PM_{2.5}$ measured by a MARGA (Allen et al., 2015) as well as predicted by a CMAQ regional chemical transport model calculation and box models for SOAS conditions at CTR. $F_p$ boxplots indicate the maximum, 75th percentile, median, 25th percentile, and minimum. Short dashes within the boxplots indicate the mean $F_p$. Box model inputs were either the aerosol (A) or aerosol and gas concentrations (A+G). Box models were run with either the ammonium-sulfate system (A') or including all cations and anions (A). AIOMFAC calculations assumed complete liquid-liquid phase separation between the organic-rich and electrolyte-rich phases (CLLPS) or employed a full equilibrium calculation with organic compounds in which phase separation was calculated based on composition (EQLB). Observed gas-phase ammonia concentrations are from the SEARCH network at CTR. Boxplots are labeled by a letter for easier reference in the text.





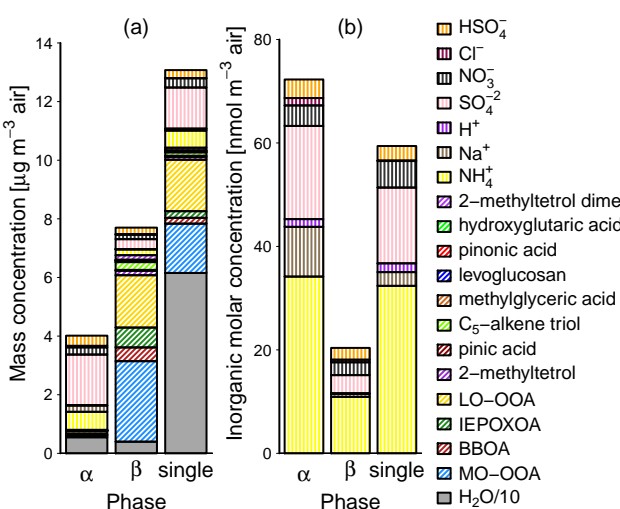

**Figure 3.** Average composition of the $\alpha$ (electrolyte-rich), $\beta$ (organic-rich), and single phase in terms of (a) mass (organic and inorganic components) and (b) moles (ions only) predicted by AIOMFAC using $PM_{2.5}$ aerosol composition observed during SOAS. Species are stacked in the same order as indicated by the legend.





**Table 1.** $[H^+]_{air}$ and pH predicted for $PM_{2.5}$ at SOAS CTR (median $\pm$ one standard deviation) under conditions of complete liquid-liquid phase separation between the organic-rich and electrolyte-rich phases (CLLPS) or in a full equilibrium calculation in which phase separation was calculated based on composition (EQLB).

| Model | CLLPS | EQLB |
|---|---|---|
| $[H^+]_{air}$ in nmol m$^{-3}$ air | | |
| AIOMFAC (A') | $1.9 \pm 1.9$ | $1.3 \pm 1.6$ |
| AIOMFAC (A) | $1.8 \pm 2.1$ | $1.1 \pm 1.8$ |
| ISORROPIA (A) | $2.0 \pm 2.8$ | NA |
| ISORROPIA (A+G) | $0.5 \pm 1.5$ | NA |
| $pH = -log_{10}(\gamma_{H+}[H^+]_{air}/[S])$ | | |
| AIOMFAC (A') | $1.3 \pm 1.2$ | $1.4 \pm 1.2$ |
| AIOMFAC (A) | $1.3 \pm 2.1$ | $1.5 \pm 2.0$ |
| ISORROPIA (A) | $0.7 \pm 2.5$ | NA |
| ISORROPIA (A+G) | $1.1 \pm 0.7$ | NA |





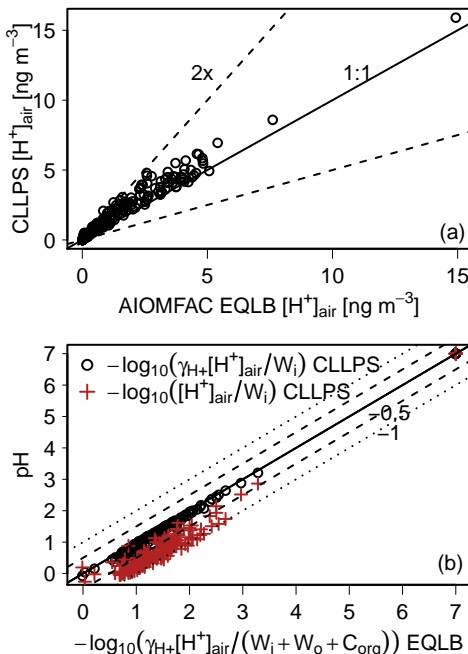

**Figure 4.** (a) $[H^+]_{air}$ and (b) pH predicted for $PM_{2.5}$ using AIOMFAC. Dashed lines in panel (a) indicate a factor of two difference from the 1:1 line. Dashed lines in (b) represent a $\pm\, 0.5$ shift in pH while dotted lines represent a $\pm\, 1$ shift in pH. Series marked in open circles ($\circ$) are summarized in Table 1. All calculations used the ammonium-sodium-sulfate-nitrate-chloride and organic compounds system.




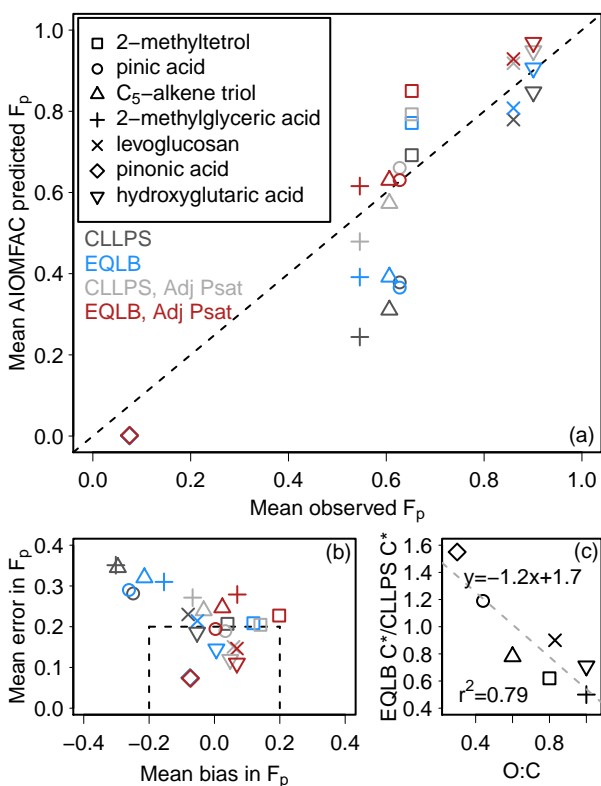

**Figure 5.** Observed and AIOMFAC-based predictions of equilibrium partitioning of organic compounds in the presence of MARGA-measured $PM_{2.5}$ inorganics. In panel (b), mean bias = $\frac{1}{n}\sum_{i=1}^{n}(M_i - O_i)$ and mean error = $\frac{1}{n}\sum_{i=1}^{n}|M_i - O_i|$ where $M_i$ is the model prediction and $O_i$ is the observation of $F_p$. The ratio of mean saturation concentration under EQLB compared to CLLPS conditions (c) uses predictions from the adjusted vapor pressure calculations (Adj Psat). Modeled particulate 2-methyltetrols are 50% dimers except with Adj Psat.





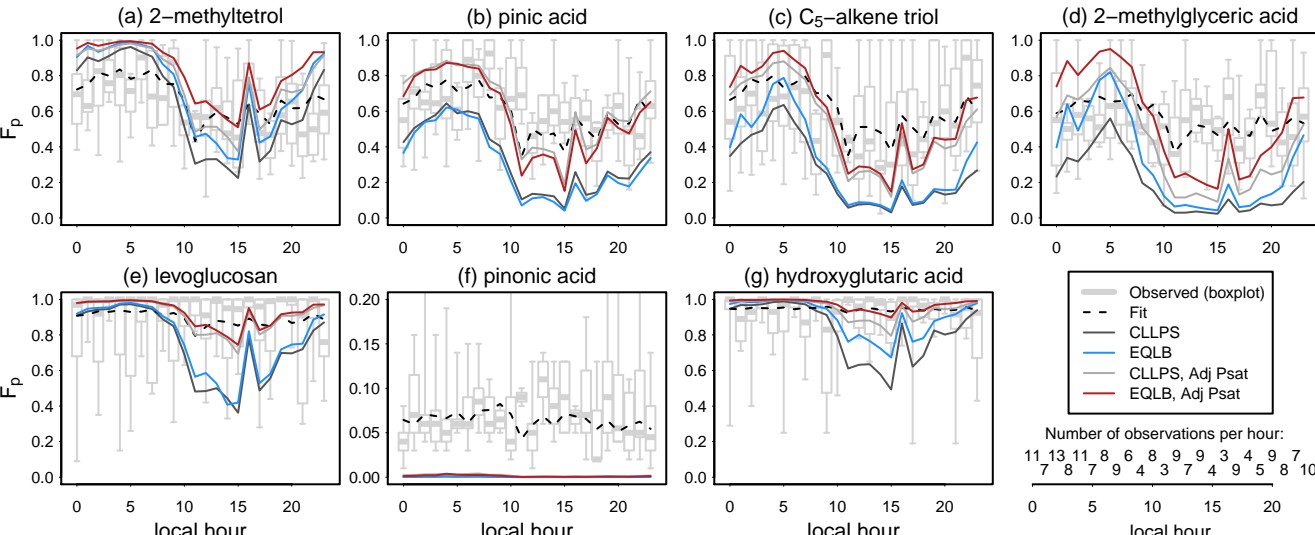

**Figure 6.** Fraction of each explicit organic species in the particle as a function of hour of day between 1 June and 15 July 2013 at CTR. 2-methyltetrols were modeled as 50% dimers in the particle for CLLPS and EQLB. When the pure species vapor pressure was adjusted, 2-methyltetrols were assumed to be entirely monomers. Fit is based on traditional absorptive partitioning to an organic compounds-only phase (Equation S1).