# Peer review of "Coupling of organic and inorganic aerosol systems and the effect on gas-particle partitioning in the southeastern United States"

_Atmospheric Chemistry and Physics, 2017_

## Referee Comment (RC1) · Anonymous Referee #1 · 13 Aug 2017

This study presented interesting results on how inorganic-organic interactions would influence particle partitioning, based on observation and simulation results from several models. Generally, this paper is comprehensive and well organized, while several concerns should be addressed before publishing.

Major Comments:

(1) In section 3.1, the authors attributed the discrepancy between CMAQ and observations to the inappropriate inclusion of cations from insoluble metal oxides. They further indicated that the overestimation of transition metals could not be avoided even if the dust emissions are closed. However, they did not state clearly whether the dust emissions were closed in their simulations, and to what extent that would make a difference. In fact, as shown in Fig. 2, CMAQ substantially underestimated ammonia Fp, while the RN/2S are comparable with other models. Does that arise from the overestimated total ammonia emissions, or more from the overestimated non-volatile cations that would bias the aerosol acidity and therefore the gas-particle partitioning? These mechanisms should be better described and quantified.

(2) In section 3.1, the different RN/2S ratios from CSN and SEARCH networks are attributed to measurement errors. However, the discrepancy is over 33%, which cannot be totally explained by the <20% measurement errors from Nylon filter. Other possible error sources should be discussed, and the influence of the observation uncertainty should also be discussed in subsequent sections.

(3) In section 3.4, when phase separation occurred, what is the acidity of each phase?

(4) In section 3.4, the authors claimed that all the high-pH points are due to measurement uncertainty, which is not convincing. Does these points all occur at very low concentrations when the uncertainty is extremely large? Moreover, they mentioned that there were some elevated nitrite episodes, probably from sea-salts. Whether the high-pH points correspond with those episodes should be examined.

(5) The implication of results shown in this study, and future work directions should be discussed more in-depth. For example, what is the major strength and weakness of current models shown in this study? Should the second organic phase of, say, HOAs, as mentioned in section 3.5, be added in the future?

Minor points:

(1)In Figure 1, the colors of 0.8~1 and 1.0~1.2 are hardly distinguishable. If this is not on purpose, please change the colormap. Also consider adding 2 panels showing the difference between observed and modeled RN/2S and R+/-.

(2) The relationship of IMPROVE and SEARCH network should be clarified or unified.

[Figure]

All through the manuscript the "SEARCH" network is referred to, while in Fig. S1 to S5 "IMPROVE" is used.

---

## Short Comment (SC1) · 21 Aug 2017

The comment was uploaded in the form of a supplement:
https://www.atmos-chem-phys-discuss.net/acp-2017-623/acp-2017-623-SC1-supplement.pdf

---

## Author Comment (AC1) · 1 Sep 2017

We thank the two anonymous reviewers for their timely comments. We will respond to their comments in a later post as we are conducting additional CMAQ simulations without cations. Here, we respond to the comment of Weber et al.

We thank Weber et al. for bringing more information to this conversation as their recently submitted ACPD paper (Guo et al. 2017) was not publically available when we prepared our paper. We agree that the choice of observational dataset is important. We assert that the role of organic compounds on inorganic aerosol partitioning is secondary to the issue of what observational dataset, particularly aerosol concentrations, is used in the analysis. Note that we used the SEARCH network measurements of gas-phase ammonia, not the MARGA. The differences in gas-phase ammonia from different techniques are summarized in Table S7 of our paper. In decreasing order of abundance: AIM $NH_3$ > MARGA $NH_3$ > CIMS $NH_3$ > SEARCH $NH_3$. By using the lowest observed $NH_3$, we give the models the best chance of success. We assert that current thermodynamic models (e.g. ISORROPIA) are adequate to first order. The impact of organics and nonvolatile cations are second order effects and do not reconcile different aerosol observational datasets. We will work to clarify that message in a revision of our paper as well as in this comment.

We do demonstrate that organic compounds have a perturbation on ammonia/ammonium partitioning and pH, mostly via the activity coefficient, which we do not assume is one in the calculation of pH. Thus, we don't think it is appropriate to shift the S-curve in Comment-Figure 3 horizontally by 0.8 as the original S-curve corresponds to an activity coefficient of 1 (Guo et al. indicate a value of 1). A more reasonable shift in the S-curve would be 0.1 to 0.2 pH units for the inclusion of organics (Table 1 including activity coefficients). Thus, there is likely no disagreement between their Comment-Fig. 3 and our model calculations with ISORROPIA or AIOMFAC as long as the pH shown on the x-axis is consistent with the definition used by the models.

We agree that nonvolatile cations play a role in particle pH and partitioning as indicated by Guo et al. (2017), however we do not think they are the driver of discrepancies currently reported in literature (e.g. Silvern et al. 2017, Weber et al. 2016). We conducted box model simulations with and without Ca, K, Mg, Na, and Cl and showed that they do have a minor effect on the resulting ratio of ammonium to 2*sulfate and ammonium vs total ammonia. Figure 2i shows ISORROPIA predictions with all available constituents input (Ca, K, Mg, Na, Cl, SO4, NH4, NO3, HNO3, NH3) while Figure 2j uses only NH4, NH3, and SO4 constituents. The resulting ratio of ammoniums to 2*sulfate and ammonium vs total ammonia is slightly affected by including the additional constituents. Their effect on the ratio of ammonium to 2*sulfate is much smaller than the difference in RN/2S derived from AMS (Figure 2a) measurements compared to MARGA (Figure 2f) measurements.

We do not use the molar ratio of ammonium to sulfate to infer acidity since we explicitly calculate pH, but we do use it as a metric for model evaluation since observations needed to calculate the value are available.

In short, we show that the results of Silvern et al. and other work (such as Weber et al. 2016) can be reconciled by recognizing that the aerosol observations themselves do not provide consistent ammonium and sulfate nor RN/2S (RN/2S=[NH4]/[2*SO4] by mol). In the examination of the role of nonvolatile cations, page 5 of Guo et al. 2017 indicates (with R=NH4/SO4):

> "predicted R was on average … $1.85 \pm 0.17$ for measured Na+ input, and the highest R at $1.97 \pm 0.02$ when zero Na+ was used as model input. The average measured R was $1.70 \pm 0.23$ by PILS-IC and $1.75 \pm 0.20$ by another PM2.5 water-soluble ion measurement (Allen et al., 2015)."

This ammonium/sulfate ratio of 1.7 from PILS is still significantly higher than the value from AMS: 0.93 (ground) or 1.21 (aircraft) in Silvern et al. or 1.2 (GT-AMS on ground) in our work. The inclusion of measured Na+ lowers R by about 0.1 (Guo et al.) while the differences in measurement-derived R for different techniques are up to 0.8. So while it is important to treat nonvolatile cations and organic compounds to fully understand the inorganic system and improve predictions of pH (and R or $R_{N/2S}$), our conclusion is that the current debate in literature over ammonium/sulfate ratios and their agreement or disagreement with models is driven by disagreement between different observational datasets.

References

Guo, H., Nenes, A., and Weber, R. J.: The underappreciated role of nonvolatile cations on aerosol ammonium-sulfate molar ratios, Atmos. Chem. Phys. Discuss., https://doi.org/10.5194/acp-2017-737, in review, 2017.

Silvern, R. F., Jacob, D. J., Kim, P. S., Marais, E. A., Turner, J. R., Campuzano-Jost, P., and Jimenez, J. L.: Inconsistency of ammonium–sulfate aerosol ratios with thermodynamic models in the eastern US: a possible role of organic aerosol, Atm. Chem. Phys., 17, 5107-5118, doi: 10.5194/acp-17-5107-2017, 2017.

Weber, R. J., Guo, H., Russell, A. G., and Nenes, A.: High aerosol acidity despite declining atmospheric sulfate concentrations over the past 15 years, Nature Geoscience, 9, 282-285, doi: 10.1038/ngeo2665, 2016.

---

## Author Response (AR1)

We thank both anonymous reviewers for their timely comments. Their comments are reproduced in black text and our responses are in blue text. Text added directly to the manuscript is *indented and italicized* or pasted via screenshot. Please see the tracked changes version as well.

**Anonymous Referee #1**

This study presented interesting results on how inorganic-organic interactions would influence particle partitioning, based on observation and simulation results from several models. Generally, this paper is comprehensive and well organized, while several concerns should be addressed before publishing.
Major Comments:

(1) In section 3.1, the authors attributed the discrepancy between CMAQ and observations to the inappropriate inclusion of cations from insoluble metal oxides. They further indicated that the overestimation of transition metals could not be avoided even if the dust emissions are closed. However, they did not state clearly whether the dust emissions were closed in their simulations, and to what extent that would make a difference. In fact, as shown in Fig. 2, CMAQ substantially underestimated ammonia Fp, while the RN/2S are comparable with other models. Does that arise from the overestimated total ammonia emissions, or more from the overestimated non-volatile cations that would bias the aerosol acidity and therefore the gas-particle partitioning? These mechanisms should be better described and quantified.

Appel et al. (2013) conducted simulations in which wind-blown dust emissions were removed from CMAQ to evaluate the effect on cation predictions. We include dust emissions following Foroutan et al. (2017) as indicated in section 2.1, and we did not repeat the sensitivity simulation of removing wind-blown dust by Appel et al. in this work. However, in response to comments from both reviewers we conducted an additional sensitivity simulation in which all nonvolatile cations (Na, K, Mg, Ca) were removed from the ISORROPIA thermodynamic calculations in CMAQ for fine aerosol. The sensitivity simulation results in RN/2S values of 1 for the eastern US demonstrating that cations cause the low RN/2S in CMAQ:

[Figure]

Figure AC1: Ratio of ammonium to 2 x sulfate predicted in a CMAQ simulation without nonvolatile cations (1 June – 15 July 2013). The model is sampled at CSN (circle) and SEARCH (triangle) locations.

Furthermore, we quantify the overestimate in nonvolatile cations in a new table in the SI and add text to the manuscript indicating the factor of 3 overestimate in nonvolatile cations. If the excess cations were corrected and replaced by ammonium, it would lead to a 26% increase in ammonium.

Revised text (abstract):

were consistent with these conditions. CMAQv5.2 regional chemical transport model predictions did not reflect these conditions due  a factor of three overestimate of the nonvolatile cations

Revised text (section 3.1):

*An overabundance of cations in the CMAQ model (Figure S1, Table S10) means that ammonium was displaced from the particle and RN/2S was biased low for the southeast U.S. An evaluation of the individual cations and anions (Figure S1, Table S10) indicated CMAQ over predicted the non-volatile ISORROPIA cations (Na+, Ca2+, K+, Mg2+) by factors of 2 to 6 individually and by a charge equivalent factor of 3 overall in the Southeast. A factor of 3 overestimate in nonvolatile cations indicates ammonium predicted by CMAQ was low by about 26%.*

*A sensitivity simulation in which all Aitken and accumulation mode Na+, Ca2+, K+, and Mg2+ were removed from the partitioning thermodynamics resulted in a mean predicted RN/2S of 0.96 for the southeast U.S.*

Underestimates in ammonia Fp are due to both the replacement of $NH_4^+$ by nonvolatile cations and to overestimates in $NH_3$ emissions (both operate in the same direction). Note that during SOAS total $NH_3$ was significantly overestimated in CMAQ (Figure S8) and removing the cations only increases NHx Fp slightly (revised Figure 2g,h). Overestimates in $NH_3$ would not cause low RN/2S.

A new boxplot "CMAQ, no cations" was added to Figure 2. Revised text:

*Note that in full CMAQ model calculations, the predicted nonvolatile cation concentrations were so high that they erroneously affected the partitioning of ammonium (Table S10, Figure S1). Removing nonvolatile cations from CMAQ (h) allowed more ammonium into the particle and led to increased RN/2S, but NHx Fp was still low indicating overestimates in gas-phase ammonia in the CMAQ model are not primarily due to the displacement of ammonium by nonvolatile cations.*

(2) In section 3.1, the different RN/2S ratios from CSN and SEARCH networks are attributed to measurement errors. However, the discrepancy is over 33%, which cannot be totally explained by the <20% measurement errors from Nylon filter. Other possible error sources should be discussed, and the influence of the observation uncertainty should also be discussed in subsequent sections.

Section 3.1 highlights some known instances of measurement bias which include: $SO_2$ collection on nylon filters and $NH_4$ volatilization from nylon filters. From Solomon et al.'s description of the IMPROVE and CSN networks: "Overall measurement accuracy is not possible for most PM measurements since

traceable standards applied in the field are not available." We have added the CSN and SEARCH measurement precision, which is much less than the difference in measurement techniques, to section 2.2:

> *Solomon et al. (2014) estimate the precision of CSN measured ammonium is 11% and sulfate is 7% (for co-located samples during 2012) but the actual measurement uncertainty is likely higher (and not quantified).*

> *SEARCH reports the precision of measured sulfate and ammonium in PM2.5 is 2-3% (Edgerton et al., 2005).*

Section 3.2 has also been revised to include a discussion of AMS uncertainty. Revised text:

> *The differences in the AMS and MARGA datasets in terms of RN/2S are larger than can be explained by known measurement precision. However, uncertainty for AMS measured ammonium (34%) and sulfate (36%) are large (Bahreini et al., 2009). A contributor to this uncertainty is the AMS collection efficiency (CE), and AMS instruments …*

(3) In section 3.4, when phase separation occurred, what is the acidity of each phase?

In a liquid-liquid phase separation case for AIOMFAC EQLB predictions, each phase had a similar pH. The predictions show a typical absolute difference of less than 0.2 pH units (frequently less than 0.1 pH units), i.e. clearly within the range of variability of the values reported in Table 1. The difference is small since liquid-liquid equilibrium thermodynamics drive activity-based pH values to have nearly the same magnitude in coexisting phases (but not necessarily exactly the same values). A sentence in section 3.3 was revised to add the H+ air value from Figure 3 to the text.

> *The β phase, with its higher concentration of organic species, showed a lower average [H+]air (0.1 nmol/m3) compared to the α phase (1.5 nmol/m3), while the activity-based pH values were predicted to be similar in both phases, typically within 0.2 pH units (as expected from equilibrium thermodynamics).*

(4) In section 3.4, the authors claimed that all the high-pH points are due to measurement uncertainty, which is not convincing. Does these points all occur at very low concentrations when the uncertainty is extremely large? Moreover, they mentioned that there were some elevated nitrite episodes, probably from sea-salts. Whether the high-pH points correspond with those episodes should be examined.

pH=7 predictions are not restricted to low concentrations (Figure AC2, left) or high NaCl concentrations. They are associated, by design, with an excess of cations compared to anions (Figure AC2, right). This is the case in the AIOMFAC-based calculations because the presence of charge-weighted excess amounts of cations in the measured ion concentration data meant that no explicit amount of $H^+$ was necessary to establish a neutral charge in the overall aerosol mixture (nor was $HSO_4^-$ quantified among the measured anions). Note that the $H^+$ concentration was not measured, but inferred via an overall charge balance constraint in cases with an excess of anions (excess negative charge) to initialize the model calculations with AIOMFAC. Therefore, by default, the pH value was set to 7 in cases where no $H^+$ ions were present in the mixture at input (autodissociation of water is not explicitly considered). Since an excess in positive charge may occur erroneously as the result of measurement uncertainties during cation and anion

quantification, it remains possible that the particles with a reported pH of 7 were in fact acidic. This hints to the challenges in accurate H⁺ concentration quantification in the field and related sensitivities in predicted particle pH. A small decrease in the cation abundance, as demonstrated by Hennigan et al. (2015) can lead to a significantly lower pH. Figure AC2 (right) shows that pH=7 is obtained when the cations meet or exceed the anions.

[Figure]

Figure AC2: (left) Predicted pH vs. molar cation abundance (ammonium+sodium+hydronium) used in AIOMFAC calculations. (right) Predicted pH vs cations-anions.

(5) The implication of results shown in this study, and future work directions should be discussed more in-depth. For example, what is the major strength and weakness of current models shown in this study? Should the second organic phase of, say, HOAs, as mentioned in section 3.5, be added in the future?

As HOA was not resolved in the AMS PMF analysis, it likely contributed <5% of total organic aerosol. A third liquid phase could be examined as part of future work.

We have added this additional text to the conclusions to emphasize our main message (also in response to the Weber et al. comment in ACPD):

> *The lack of agreement of AMS and CSN data with thermodynamic models, but the agreement between MARGA observations and models, indicates a potential bias in CSN measurements and that AMS data alone may not be suitable for thermodynamic modeling. The diversity in observational datasets can explain why some work has concluded thermodynamic models fail (Silvern et al., 2017) while others indicate models are adequate (Weber et al., 2016). This work finds thermodynamic…*

To highlight the strength of the AIOMFAC-based model, we add in Section 3.2:

> *Comparing the change in mean NHx Fp with (m) and without (l) organic compound interactions indicates that organic compounds have a larger effect on ammonia gas-particle partitioning than the inclusion (j) or lack (k) of calcium, potassium, magnesium, sodium, nitrate, and chloride.*

**Minor points:**
(1)In Figure 1, the colors of 0.8~1 and 1.0~1.2 are hardly distinguishable. If this is not on purpose, please change the colormap. Also consider adding 2 panels showing the

difference between observed and modeled RN/2S and R+/-.

We have changed the color bar. A new table, Table S10, provides CMAQ model performance metrics vs CSN and SEARCH for all cations and anions involved in ISORROPIA calculations.

(2) The relationship of IMPROVE and SEARCH network should be clarified or unified. All through the manuscript the "SEARCH" network is referred to, while in Fig. S1 to S5 "IMPROVE" is used.

SEARCH and IMPROVE are different networks. The IMROVE network data does not appear in the main manuscript since it does not measure ammonium. We added the following to section 2.3:

> *The Interagency Monitoring of Protected Visual Environments (IMPROVE) network (Solomon et al. 2014) also measures some chemical speciation of PM2.5 throughout the U.S., but does not include ammonium.*

> *The SEARCH network operates at fewer sites and exclusively in the Southeast U.S.*

**Anonymous Referee #2**

**General Comments**
This study examines the partitioning of inorganic and organic species in the southeast US summer using information from a large number of observational data sets and explores several different modeling frameworks to describe the partitioning of these species with varied levels of sophistication and agreement with observations. The inclusion of organics and phase separation as well as the organic compound composition is shown to be important for determining aerosol acidity and ammonia partitioning. This work is an important contribution to the modeling community's treatment of inorganic and organic aerosol species. The paper is well written and highly detailed, and below are specific comments to be considered in the revision of this manuscript.

**Specific Comments**
Page 3, lines 1-3: NAAQS are set for certain particulate precursors (SO2, NO2), so is the reference specifically to NH3 and organic gases?

Yes. *($NH_3$ and organic compound vapors)* has been added.

Page 3, line 18: Please cite NEI version used in CMAQ simulations and clarify "specific when available."

2011 National Emission Inventory version 2 is already indicated, but we have added "ek" as further identification. We also revised "specific when available" to:

> *Emissions influenced by model meteorology (biogenic compounds, mobile sector) or monitored (electrical generation units) were year 2013 specific.*

Page 3, lines 27-29: It would be helpful for the authors to distinguish between the use

of forward and reverse modes of ISORROPIA for clarity and consistency with thermodynamic modeling literature.

Forward and reverse mode definitions have been added to the text in section 2.1.

Section 2.2: How do the AMS measurements used in this study compare to the other AMS measurements made at the same site during the measurement campaign (Hu et al., 2015), which do include nitrate (and chloride). How do SOAS measurements compare with SENEX and SEAC4RS measurements?

An examination of SENEX data is beyond the scope of our paper, but has been modeled in previous work. All of the AMS measurements from SOAS and SEAC4RS are similar. Table S5 shows a comparison of SOAS-CTR measurements while table S6 summarizes results from SEAC4RS and the work of Silvern et al (2017). All AMS data indicate average ratios of ammonium to 2*sulfate of 0.6 or lower. Nitrate measured by the AMS includes both inorganic (able to participate in inorganic aerosol thermodynamics) and organic (Xu et al. 2015b) compounds. Thus we do not consider nitrate in our AMS-based thermodynamic calculations since the fraction that is inorganic is not precisely known. Running ISORROPIA with and without Ca, K, Mg, Na, Cl, and NO3 from MARGA (Figure 2i,j) indicates they are not present in enough abundance to significantly change ammonium to sulfate ratios or the fraction of NHx in the particle (i.e. the ratios change by less than 0.1). Added to section 2.2:

> This AMS dataset was consistent with the other AMS instrument operating at CTR as well as AMS measurements aboard an aircraft over the southeastern U.S. (Table S5-S6).

Page 5, line 32-page 6, line 1: Clarify the trigger condition; what is the ammonium associated with if NH4+ ≥ 2 × SO42- ?

Measured concentrations have uncertainty and thus NH4+ ≥ 2 × SO42- could represent unrealistic conditions. Minor amounts of NH4 could be associated with Cl or NO3. The reason for introducing bisulfate is a technical one; it is done since for a $H^+$ and bisulfate-free electrolyte input, i.e. only $(NH_4)_2SO_4$ at input (no $NH_4HSO_4$ nor $H_2SO_4$), no explicit acidity nor partial association of $H^+$ and $SO_4^{2-}$ to $HSO_4^-$ are considered by the AIOMFAC model. Therefore, such an input would always lead to a neutral (pH = 7) solution. However, the addition of a tiny amount of $NH_4HSO_4$ triggers the bisulfate dissociation equilibrium calculation and the pertaining acidity and pH-dependent ammonia gas-particle partitioning under given conditions. The additional AIOMFAC-based partitioning calculations using all anion amounts (Cl, NO3, SO4, NH4, equivalent Na) did not perform the resetting of mass (i.e introduction of bisulfate). We clarified that the introduction of bisulfate was performed for MARGA sulfate-ammonium (A') inputs only. Since AMS data never met this condition, the location of the sentence was also moved to after the MARGA data was introduced and additional text added:

> The data from MARGA was used in two ways for model calculations with AIOMFAC. (1) All the measured ion concentrations were considered, but the molar amounts of the cations Ca2+, K+, Mg2+, and Na+ were mapped to a charge-equivalent amount of Na+ (see Section 2.1). (2) Only the measured concentrations of ammonium and sulfate ions were considered and mapped to the electrolyte components ammonium sulfate, ammonium bisulfate, and sulfuric acid for AIOMFAC model input purposes. For ammonium-sulfate only conditions (option 2) in which the moles $NH_4^+ > 2xSO_4^{2-}$, a small amount ($1 \times 10^{-4}$ µg m$^{-3}$) of ammonium bisulfate was added to the AIOMFAC input for MARGA calculations in order to trigger a potential partial association of

sulfate and H+ ions to bisulfate following the equilibrium constant of that reaction. *Such conditions did not occur with AMS data.*

Section 2.2 SEARCH ammonia observations: Are these measurements hourly or 24-hr averages and are they only coincident with SOAS AMS observations? Are the measurements sensitive to diurnal variations and possible measurement interferences at cold temperatures/low relative humidity?

Box model calculations use hourly inputs of ammonia. Very low RH and cold temperatures were likely not experienced during summer at SOAS. We add:

> *Hourly* Gas-phase ammonia was obtained from the CTR SEARCH network site via a corrected Thermo Scientific citric acid-impregnated denuder.

Section 3.1: It would be helpful to include a discussion of model biases in ammonium and sulfate, even if RN/2S is okay. Why is there a disagreement in the model bias between IMPROVE and CSN sites for sulfate? Is sea salt sulfate an important contributor to total sulfate in the model and observations along the coasts?

We do not specifically examine the role of sea salt mediated sulfate or sites on the coast, but that topic has been examined in other work (Kelly et al., 2010 GMD https://www.geosci-model-dev.net/3/257/2010/ for example). Na and Cl are not major drivers of ammonium to sulfate ratios as indicated by sensitivity simulations conducted with and without Na, Cl, K, Mg, Ca, and NO3 for SOAS.

IMPROVE sites specifically target Class I areas such as National Parks and Wilderness areas. The overestimate in CMAQ-predicted sulfate compared to CSN, but underestimate compared to IMPROVE could indicate a bias as a function of photochemical age in the model, but sulfate is relatively unbiased overall (normalized mean bias of 5% compared to CSN and -12% compared to SEARCH in the Southeast, Table S10). New text added:

> *CMAQ-predicted sulfate was relatively unbiased in the southeastern U.S. (normalized mean bias of 5% compared to CSN), but ammonium was high by a factor of 1.5 (Table S10).*

We have refocused the supporting information on sulfate and ammonium as well as CSN and SEARCH:
Figure 1b was eliminated since the Central US is not a focus of the study.
Figure S4 showing the nitrate bias has been eliminated since nitrate is not a major constituent affecting inorganic PM in the southeast US in summer.
Figure S5 was eliminated. The main goal of Figure S5 was to show ammonium and sulfate which is contained in other plots.
Figure S2-S3: The included data sets have been expanded to include the SEARCH network.

A table of model performance for ammonium and sulfate (as well as other species) has been added (Table S10). For additional evaluation of sulfate and ammonium during other seasons and across the U.S., we refer the reader to Appel et al. 2017 (https://www.geosci-model-dev.net/10/1703/2017/gmd-10-1703-2017.pdf).

Section 3.1: It is unclear why Na, Ca, K, and Mg used in ISORROPIA's calculations in CMAQ shouldn't be. If the problem is that their concentrations in CMAQ are overesti

mated as is pointed out in the text, what can be done? It is a concerning conclusion to draw that CMAQ can't be used to diagnose species partitioning.

CMAQ still provides valuable information and we demonstrate via box modeling that if the nonvolatile cation abundance can be corrected, the thermodynamics based on ISORROPIA should operate consistently with how MARGA and SEARCH observations indicate the system should behave. We now include an estimate of the cation overabundance (factor of 3 overall, factor of 2-6 for individual cations, Table S10) as a benchmark for resolving emission issues.

We also add:

> By performing ISORROPIA and AIOMFAC box modeling, this work demonstrates that our current thermodynamic understanding of ammonium and sulfate aerosol is consistent with (MARGA) observations in the southeastern U.S. atmosphere. Since models like CMAQ use the same thermodynamic basis, specifically ISORROPIA, these results build confidence that regional models can capture the thermodynamics of the ambient atmosphere. However, our results also demonstrate that for the partitioning of ammonia and ammonium to be correct, errors in emissions of nonvolatile cations, on the order of a factor of 3, must be resolved as well.

Figures S2-S4: It would be helpful to add SEARCH observations to these plots as a supplement to the ratios shown in Figure 1.

Added

Page 7, lines 22-23: Why a 10% overestimate in sulfate if in line 16 there is a 4-5% overestimate cited?

The 4-5% overestimate was due to $SO_2$ uptake onto filters in studies from the 1990s; as stated in the manuscript, it refers to $4-5\%$ uptake of gas phase $SO_2$ onto nylon filters, which is not to be interpreted as a $4-5\%$ overestimate in particulate sulfate. We do not know the magnitude of the particulate sulfate artifact for conditions in the current SE US. We point out that only a 10 % overestimate in sulfate combined with a 10 % underestimate in $NH_4$ could have the order of magnitude effects on RN/2S that we observe.

Page 7, line 31-page 8, line 1: Why is NHx Fp so similar and RN/2S very different between the two ISORROPIA runs?

This comment refers to ISORROPIA simulation of AMS data using aerosol-only (forced to reproduce RN/2S) or gas + aerosol inputs (RN/2S is output). If the observed system is consistent with the thermodynamics represented in ISORROPIA, RN/2S and NHx Fp should not be drastically different in aerosol-only and gas + aerosol mode calculations (see additional lines in Figure 2 and caption). The fact that the predicted RN/2S is very different, indicates that the gas + aerosol input system is inconsistent with the model-predicted ammonia partitioning when only aerosol composition is provided as input. We indicate that one simulation is forced to reproduce RN/2S:

> ISORROPIA predictions using AMS measured ammonium and sulfate as input (c), *thus exactly reproducing observed R{N/2S}*, showed much higher partitioning of ammonia to the particle phase...

Page 8, lines 12-14: Visually it doesn't look like NHx Fp for the two CMAQ particles sizes looks very different. Stating values in the text could help make that point clearer.

The values are not meaningfully different.

Page 8, lines 15-17: If ISORROPIA was not sensitive between two different particle sizes, why was CMAQ?

CMAQ uses a modal size distribution. To determine PM1 and PM2.5 the amount of mass below 1 and 2.5 microns respectively is obtained from the Aitken+Accumulation modes. Thus the absolute abundance of ammonium and sulfate is slightly different for PM1 compared to PM2.5, but the ratio of ammonium to sulfate is not different.

Page 8, lines 20-21: Could a full CMAQ model run with only NH4, SO4, NO3, Cl included in the thermodynamic calculations be done?

The simulation was performed (see response to Reviewer 1). Removing K, Mg, Ca, and Na resulted in CMAQ predicted RN/2S values of 0.96 averaged across the southeastern US CSN sites.

Figure 2: This figure is slightly difficult to interpret given the large amount of information concentrated in it. What significance does the different gray shading have for the box plots – that should be defined in the figure caption. NHx could be shown on a separate panel to separate some of the information.

The figure was created to compactly show several pieces of information in a limited amount of space. Horizontal lines have been added to help guide the eye. Shading is to visually group different simulations (CMAQ vs ISORROPIA vs AIOMFAC vs observations). Lines have been added to guide the eye and caption updated:

[Figure]

**Figure 2.** Gas-particle partitioning of ammonia ($NH_x$ $F_p$ = ammonium/(ammonia+ammonium)), mean $R_{N/2S}$ (red ×), and mean $R_{+/-}$ (blue ○) for $PM_1$ measured by the Georgia Tech AMS (Xu et al., 2015a) and $PM_{2.5}$ measured by a MARGA (Allen et al., 2015) as well as predicted by a CMAQ regional chemical transport model calculation and box models for SOAS conditions at CTR. $F_p$ boxplots indicate the maximum, 75th percentile, median, 25th percentile, and minimum. Short dashes within the boxplots indicate the mean $F_p$. Box model inputs were either the aerosol (A) or aerosol and gas concentrations (A+G). Box models were run with either the ammonium-sulfate system (A') or including all cations and anions (A). AIOMFAC calculations assumed complete liquid-liquid phase separation between the organic-rich and electrolyte-rich phases (CLLPS) or employed a full equilibrium calculation with organic compounds in which phase separation was calculated based on composition (EQLB). Observed gas-phase ammonia concentrations are from the SEARCH network at CTR. Boxplots are labeled by a letter for easier reference in the text. Shading of the boxplot interquartile range distinguishes different models (CMAQ, ISORROPIA, and AIOMFAC). The horizontal lines correspond to mean observed $NH_x$ $F_p$ (black) and $R_{N/2S}$ (red). A simulation is consistent with observations if it reproduces both $NH_x$ $F_p$ and $R_{N/2S}$.

Page 8, lines 31-32: If organosulfates were not present in large enough concentrations to explain a 20% bias, are there any other explanations? Also, should cite other organosulfate measurements from SOAS (Budisulistiorini et al., 2015; Hettiyadura et al., 2017).

Citations were added. The presence of organosulfates is one hypothesis. If there is an error in how the collection efficiency and adjustment of AMS data is performed, that could be another error. The AMS has a collection efficiency of approximately 50% and thus AMS derived data is corrected to compensate for that.

> *Furthermore, organosulfates (Budisulistiorini et al., 2015; Hettiyadura et al., 2017) can be measured in the AMS as sulfate. However, organosulfates have been estimated to account for only 5% of AMS-measured sulfate during SOAS (Hu et al., 2017).*

Page 9, lines 17-20: What is the reason for the differences between this work and Pye et al. (2017), and which work compares better to observations in terms of expected frequency of phase separation?

The two approaches use different methods. The 2017 paper used an empirical representation of separation relative humidities based on O:C (or OM/OC) without knowledge of the specific organic

aerosol chemical constituents. Observations of phase separation are not available for model comparison. We added "*empirical*" to the description of the previous modeling.

Section 3.4: In discussion of aerosol pH, are there notable differences in aerosol liquid water content between models that would further impact the pH calculations?

This is possible, but we find that AIOMFAC reproduces the ISORROPIA mean pH of 0.7 if operated in the same manner (ie CLLPS mode, assumption of activity coefficient for H+ of 1).

Page 12, lines 9-11: The discussion of improvements required for CMAQ is not sufficient, especially because those errors prevented the use of the full CMAQ results in this study. If the current CMAQ thermodynamic partitioning is not usable, it is important to note if so and why.

We have revised the conclusions (see earlier responses) to highlight that we do not find motivation to change the thermodynamic basis of CMAQ, but need to resolve nonvolatile cation abundance.

**Technical corrections**
Eastern should not be capitalized on page 2 line 28 for consistency with the rest of the text and US should be U.S. on page 8 line 6.

Corrected

A few unnecessary sentences that distracted from the message of the paper were removed. From conclusions.
Removed since we didn't examine diffusivity limitations ourselves:

[revised manuscript text omitted]
 | $\gamma$ CLLPS $\beta$ phase | $\gamma$ EQLB $\beta$ phase | $\gamma$ EQLB $\alpha$ phase | Ratio: $\gamma_\beta$ EQLB/$\gamma_\beta$ CLLPS |
|---|---|---|---|---|
| 2-methyltetrol | 0.63 | 0.77 | 4.7E+03 | 1.23 |
| pinic acid | 5.22 | 16.21 | 1.4E+09 | 3.10 |
| C$_5$-alkene triol | 1.37 | 2.04 | 9.3E+04 | 1.49 |
| 2-methylglyceric acid | 0.49 | 0.48 | 23 | 0.97 |
| levoglucosan | 0.42 | 1.02 | 1.4E+05 | 2.45 |
| pinonic acid | 26.60 | 121.31 | 1.3E+10 | 4.56 |
| hydroxyglutaric acid | 0.36 | 0.96 | 290 | 2.63 |

Table S10: Comparison of CMAO predicted aerosol species concentrations and CSN and SEARCH network observations in the Southeast U.S. NOAA climate region.

| Species | Network | n | Mean $O_i$ ($\frac{\mu g}{m^3}$) | Mean $O_i$ ($\frac{\mu mol}{m^3}$) | Mean $M_i$ ($\frac{\mu g}{m^3}$) | Mean $M_i$ ($\frac{\mu mol}{m^3}$) | Mean($M_i$)/Mean($O_i$) | $r^2$ | MB ($\frac{\mu g}{m^3}$) | ME ($\frac{\mu g}{m^3}$) | NMB (%) | NME (%) | FB (%) | FE (%) | IofA | RMSE ($\frac{\mu g}{m^3}$) |
|---|---|---|---|---|---|---|---|---|---|---|---|---|---|---|---|---|
| $SO_4^{-2}$ | CSN | 225 | 1.53 | 0.0159 | 1.61 | 0.0168 | 1.1 | 0.48 | 0.07 | 0.46 | 5 | 30 | 4 | 32 | 0.82 | 0.59 |
| $SO_4^{-2}$ | SEARCH | 97 | 1.77 | 0.0184 | 1.56 | 0.0163 | 0.9 | 0.49 | -0.22 | 0.50 | -12 | 28 | -16 | 35 | 0.82 | 0.65 |
| $NH_4^+$ | CSN | 225 | 0.27 | 0.0147 | 0.40 | 0.0221 | 1.5 | 0.50 | 0.13 | 0.20 | 50 | 75 | 44 | 75 | 0.79 | 0.26 |
| $NH_4^+$ | SEARCH | 95 | 0.54 | 0.0301 | 0.38 | 0.0211 | 0.7 | 0.37 | -0.16 | 0.26 | -30 | 47 | -57 | 71 | 0.71 | 0.31 |
| $Na^+$ | CSN | 224 | 0.05 | 0.0024 | 0.12 | 0.0053 | 2.3 | 0.36 | 0.07 | 0.08 | 126 | 146 | 72 | 88 | 0.62 | 0.12 |
| $Na^+$ | SEARCH | 93 | 0.05 | 0.0020 | 0.12 | 0.0053 | 2.6 | 0.49 | 0.08 | 0.08 | 164 | 168 | 74 | 80 | 0.49 | 0.14 |
| $Ca^{+2}$ | CSN | 224 | 0.03 | 0.0007 | 0.13 | 0.0032 | 4.8 | 0.45 | 0.10 | 0.11 | 378 | 387 | 118 | 122 | 0.36 | 0.17 |
| $Ca^{+2}$ | SEARCH | 201 | 0.03 | 0.0007 | 0.16 | 0.0040 | 5.9 | 0.67 | 0.13 | 0.13 | 490 | 490 | 118 | 118 | 0.23 | 0.24 |
| $Mg^{+2}$ | CSN | 224 | 0.00 | 0.0002 | 0.02 | 0.0007 | 4.1 | 0.04 | 0.01 | 0.02 | 308 | 379 | 76 | 101 | 0.38 | 0.02 |
| $K^+$ | CSN | 224 | 0.06 | 0.0016 | 0.17 | 0.0042 | 2.7 | 0.09 | 0.10 | 0.12 | 166 | 186 | 57 | 77 | 0.24 | 0.18 |
| $K^+$ | SEARCH | 201 | 0.06 | 0.0015 | 0.18 | 0.0045 | 3.1 | 0.05 | 0.12 | 0.13 | 207 | 228 | 61 | 78 | 0.18 | 0.22 |
| $Cl^-$ | CSN | 224 | 0.02 | 0.0005 | 0.03 | 0.0009 | 1.6 | 0.09 | 0.01 | 0.03 | 61 | 148 | 17 | 97 | 0.46 | 0.07 |
| $NO_3^-$ | CSN | 225 | 0.31 | 0.0049 | 0.27 | 0.0044 | 0.9 | 0.07 | -0.04 | 0.22 | -12 | 73 | -48 | 84 | 0.43 | 0.35 |
| $NO_3^-$ | SEARCH | 97 | 0.05 | 0.0008 | 0.30 | 0.0048 | 6.0 | 0.18 | 0.25 | 0.27 | 497 | 542 | 48 | 110 | 0.10 | 0.62 |

For a given set of $n$ model predictions, $\{M_i\}$, and observations, $\{O_i\}$:

MB, Mean bias $= \frac{1}{n}\sum_1^n (M_i - O_i)$

ME, Mean error $= \frac{1}{n}\sum_1^n |M_i - O_i|$

NMB, Normalized mean bias $= \frac{\sum_1^n (M_i - O_i)}{\sum_i^n O_i} \times 100\%$

NME, Normalized mean error $= \frac{\sum_1^n |M_i - O_i|}{\sum_i^n O_i} \times 100\%$

FB, Fractional bias $= \frac{1}{n}\frac{\sum_1^n (M_i - O_i)}{\sum_i^n (M_i + O_i)/2} \times 100\%$

FE, Fractional error $= \frac{1}{n}\frac{\sum_1^n |M_i - O_i|}{\sum_i^n (M_i + O_i)/2} \times 100\%$

IofA, Index of agreement $= 1 - \frac{\sum_i^n (O_i - M_i)^2}{\sum_i^n (|M_i - \bar{O}| + |O_i - \bar{O}|)^2}$

RMSE, Root mean square error $= \sqrt{\frac{\sum_1^n (M_i - O_i)^2}{n}}$

Figure S1: Observed (CSN, IMPROVE) and modeled (CMAQ) ions for June 1, 2013 to July 15, 2013.

(a) Major cations and anions for the Southeast USU.S. NOAA Climate Region (FL, GA, SC, NA, VA)

[Figure]

(b) Major cations and anions for the Central US NOAA Climate Region

[Figure]

Figure S2: Observed (CSN-circle, SEARCH-triangle) and modeled (CMAQ) ammonium for June 1, 2013 to July 15, 2013. Ammonium is not measured by the IMPROVE network.

(a) Observed Ammonium (μg m⁻³)

[Figure]

(b) Modeled – Observed Ammonium (μg m⁻³)

[Figure]

Figure S3: Observed (IMPROVE-square, CSN-circle, SEARCH-triangle) and modeled (CMAQ) sulfate for June 1, 2013 to July 15, 2013.

(a) Observed sulfate (µg m$^{-3}$)

[Figure]

(b) Modeled – Observed sulfate (µg m$^{-3}$)

[Figure]

units = ug/m3
coverage limit = 75%

TRIANGLE=IMPROVE; CIRCLE=CSN;

[revised manuscript text omitted]

---

## Author Response (AR2)

Please note the following minor change were made compared to the accepted version:

1. Added "Department of Biomedical Engineering and Mechanics, Virginia Tech, Blacksburg, Virginia, USA" affiliation to Hosein Foroutan and adjusted numbering

2. Removed this sentence at the request of a coauthor as the PILS measured nonvolatile cations are not universally lower than all measurement techniques:

 ", the levels of nonvolatile cations would need to be larger than current measurements indicate (Guo et al., 2017). Furthermore,"

3. Updated the number of pages, figures, and tables on the SI cover page

4. Removed phase "if accepted" in context of data availability